# Clinoptilolite—An Efficient Carrier for Catalytically Active Nano Oxide Particles

Jelena Pavlović *[ID] and Nevenka Rajić

Innovation Center of Faculty of Technology and Metallurgy, University of Belgrade, Karnegijeva 4, 11120 Belgrade, Serbia; nena@tmf.bg.ac.rs
* Correspondence: jelena.pavlovic@tmf.bg.ac.rs

**Abstract:** Many efforts have been devoted to produce green materials and technology to prevent and minimize the adverse effects on the environment and human health caused primarily by population growth and industrial progress. Over the past years, the use of zeolites has attracted considerable attention as both an economically and environmentally friendly option. Zeolites are crystalline, hydrated aluminosilicates with an open-framework structure. Unique structural features make them very useful ion-changers, adsorbents and catalysts. The catalytic use of zeolites has expanded from traditional use in the petrochemical industry and refineries to use in the catalytic degradation of various environmental pollutants and the synthesis of fine chemicals. In recent times, progress on the use of zeolites has been achieved in biomass conversion to fuels and valuable industrial bio-based chemicals. This review highlights the recent advances in the catalytic application of clinoptilolite (CLI), the most abundant and explored natural zeolite. The main goal of the review is to give the current state of CLI applications and insights into CLI catalytic performance, which opens possibilities for a variety of applications.

**Keywords:** clinoptilolite; natural zeolite; catalysis; photodegradation; biomass conversion; environment chemistry; sustainability

## 1. Introduction

Today, environmental pollution is one of the most important topics of worldwide discussions. An increase in the population along with rapid industrialization and urbanization have led to damage of the environment and consequently to serious damage to human health. Furthermore, the situation is becoming more complex due to insufficient attention and control of the discharging of many pollutants to the environment, such as heavy metals, pharmaceuticals, pesticides, organic dyes, etc. Many of these pollutants are persistent, toxic and carcinogenic. Also, the generation of large amounts of solid waste and its inappropriate disposal negatively affect the environment.

Consequently, research efforts are focused on developing new materials and technologies that can minimize environmental pollution. According to the principles of sustainable development, they are necessary to be not only effective but also environmentally and economically acceptable. In this regard, catalysis is one of the main fields that can give a significant contribution to the field of green chemistry and environmental protection. It plays a key role in many chemical reactions due to the ability to selectively accelerate only the main reaction until equilibrium is reached. About 90% of all industrial processes are catalyzed, so the choice of catalyst is one of the key parameters for the sustainability of applied technologies. A large number of both homogeneous and heterogeneous catalysts are in use. Due to the problems with catalyst separation, its reusability and usually negative environmental impact, homogeneous catalysis is less preferable than heterogeneous ones. However, heterogeneous catalysts can also have their drawbacks due to poor selectivity and problems with regeneration that reduce their efficiency. Thus, the development of

catalysts that are more active, selective and stable is still a great challenge. Different kinds of solids, including activated carbon-based materials, mesoporous silica, clays, zeolites and zeolite-like materials have been studied in catalysis [1–7].

Zeolites are hydrated aluminosilicates with unique structural features and a chemical composition that is applicable in many areas. In the second part of the 20th century, there was an expansion of synthetic zeolites with new structural features, which led to the neglect of natural zeolites in many scientific studies. However, the 21st century, as the century of green chemistry, brought natural zeolites back into the spotlight of scientific interest. Many regions in the world have deposits with high contents of zeolites with high purity. Due to their low price and availability, they have become a good basis for the development of new green catalysts.

This review summarizes the results of representative studies on the catalytic activity of clinoptilolite (CLI), the most abundant and explored natural zeolite. A brief overview of the structural features of CLI is given, as well as the methods by which the catalytic performance can be improved or new catalysts can be obtained. The focus of the review is the catalytic activity of CLI in the following: (a) photodegradation of organic dyes, (b) degradation of pharmaceuticals, (c) biomass conversion to biofuels and bio-based chemicals, and (d) conversion of NOx.

## 2. A Brief Description of Zeolite Structures

Zeolites are crystalline, open-framework aluminosilicates. Their three-dimensional framework consists of tetrahedral $[SiO_4]^{4-}$ and $[AlO_4]^{5-}$ units linked via oxygen atoms (Figure 1). The zeolite lattice is negatively charged, and the metal cations, usually alkaline and earth alkaline situating inside channels and cavities, provide electroneutrality to the lattice. These cations interact with the lattice via electrostatic interactions and are movable, which enables zeolites to act as ion exchangers.

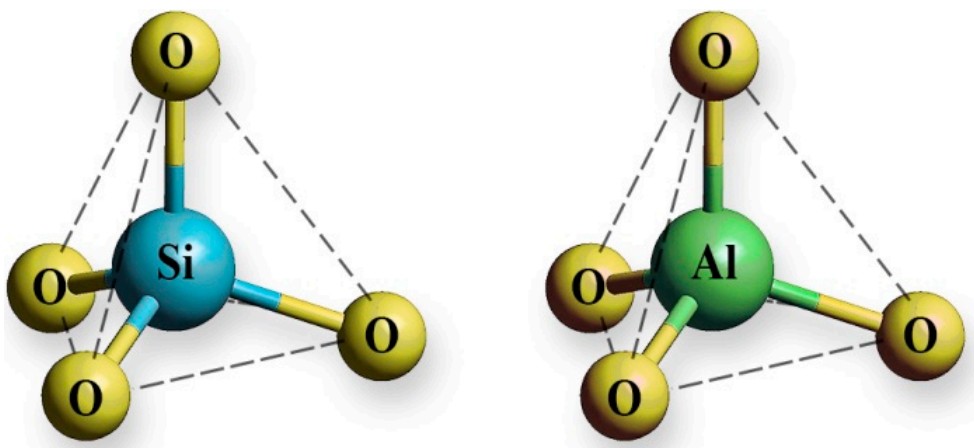

**Figure 1.** $[SiO_4]^{4-}$ and $[AlO_4]^{5-}$ tetrahedral units of a zeolite lattice.

Porosity of the lattice provides a large specific area, being several hundred to thousand square meters per gram, which gives zeolites good adsorptive properties. The shape and openings of cavities and channels affect the adsorptive behavior allowing only species of proper dimensions and geometries to enter and diffuse throughout the lattice.

Zeolites show also catalytic properties. Hydrogen ions can be accommodated in the zeolite lattice instead of metal cations, which brings Brönsted acidity and makes zeolites catalytically active. The presence of three-coordinated Al and/or extra-framework Al species in the lattice brings Lewis acid sites (Figure 2). The acidity of zeolites can be markedly enriched through different chemical and/or thermal treatments.

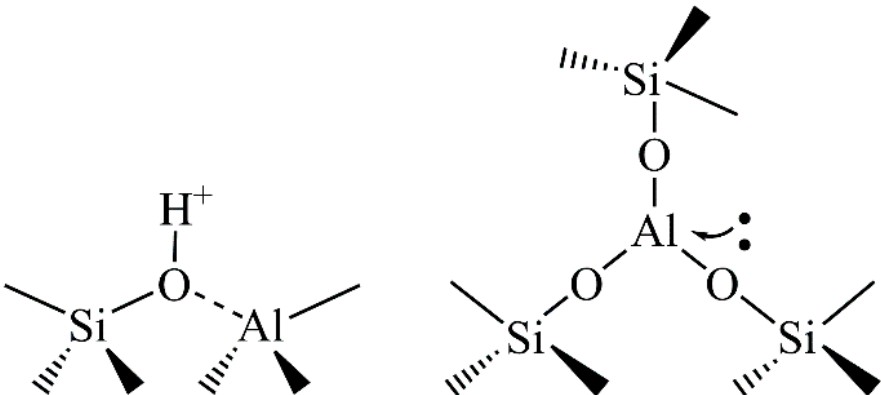

**Figure 2.** The Brönsted (**left**) and Lewis (**right**) acid sites in a zeolite lattice.

### 2.1. Clinoptilolite

Currently, about 60 different natural zeolite structures and 248 synthetic ones have been reported by the International Zeolite Association (IZA) [8]. Among the natural zeolites, CLI is one of the most abundant and the most widely studied natural zeolites. CLI-rich tuffs are located around the world, including deposits in China, the United States, Indonesia, New Zealand, Cuba and the Republic of Korea. In Europe, the abundant deposits are found in Turkey, Hungary, Slovenia, Slovakia, Ukraine, Italy, Romania and Serbia [9–11]. Depending on the location of deposits, the CLI content varies between 60% and 90%, whereas feldspars, clays and quartz are the most common present satellite phases.

CLI is a member of the heulandite (HEU) group of natural zeolites. CLI and HEU are isostructural and differ only by the Si/Al molar ratio, which influences their thermal stability. The Si/Al ratio of CLI is in the range 4.0–5.3, in contrast to HEU, for which this ratio is lower than 4.0 [9,12,13]. A higher Si/Al ratio of CLI makes CLI more thermally stable (up to 800 °C) than HEU (up to 550 °C).

The framework of CLI consists of three types of channels (Figure 3). Two types are parallel to each other, denoted as type A—formed from 10-membered rings (0.3 × 0.76 nm)—and type B—formed from 8-membered rings (0.33 × 0.46 nm). Type C consists of 8-membered rings (0.26 × 0.47 nm) and intersects the channels A and B [14,15].

(a)             (b)

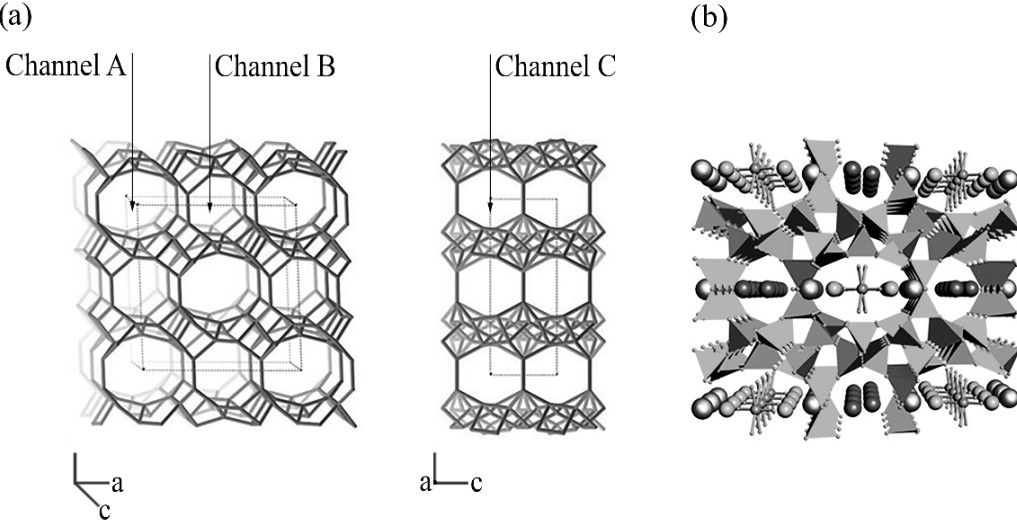

**Figure 3.** Schematic view: (**a**) 3D-structure of CLI lattice (International Zeolite Association, IZA Structure Commission, (accessed on 13 May 2023) and (**b**) cavities and channels inside the CLI lattice (tetrahedra represent the $[SiO_4]^{4-}$ and $[AlO_4]^{5-}$ building units); balls show ion-exchangeable cations [16].

### 2.1.1. Modification of CLI

In order to expand the range of CLI applications, the modification of CLI by various chemical and/or thermal treatments has received much attention in many research works.

Numerous studies have shown that the ion-exchange capacity of CLI can be significantly increased through modification. The most commonly used method involves the conversion of parent CLI into a homoionic form usually by treating CLI with a concentrated NaCl solution [17–20]. This method involves an ion-exchange reaction between cations present in the CLI lattice and cations in solution, as well as diffusion of the outside cations through the lattice. Ion exchange mediated by transition metal cations typically includes the chemisorption or complexation of the cations and/or oxide precipitation onto the CLI surface.

The surface charge and polarity of CLI can be adjusted using suitable methods that involve the adsorption of various anions and/or nonpolar organics. Among the most commonly studied methods is the adsorption of different surfactants, such as quaternary ammonium compounds (tetraethylammonium, hexadecyltrimetylammonium, cetylpyridinium or octadecyltrimethylammonium ion) [21,22]. One or two surfactant layers cover the surface depending on the surfactant concentration.

In addition, different organic species, such as polymers (polypyrrole, polyaniline, polydomin, chitosan, polyethylenimine, etc.) or amines (n-octadecylamine, n-butylamine, tetrapropylamine, monoethanolamine, 1-dodecylamine, 1-hexadecylamine, etc.) have been applied to provide the presence of multifunctional groups onto the CLI surface and additional binding sites [23–27]. Various studies confirmed that the catalytic performance of CLI can be significantly improved by covering the CLI surface with various metal oxide particles, such as Ni, Ga, Ti, Sn and Zr. The CLI structure fulfills multiple roles: it contributes to the crystallization of nano particles, prevents their agglomeration and efficiently immobilizes them [28–31].

### 2.1.2. Preparation of CLI-Based Catalysts

One of the most commonly used methods includes the conversion of CLI into a Fe(III)-form: $Fe_2O_3$-containing CLI (MOCLI). The methods are based on a three-step procedure: (1) conversion of CLI to NaCLI, (2) conversion of NaCLI to FeCLI accompanied by the addition of NaOH and (3) calcination of FeCLI at about 550 °C [32,33]. In the last step, nano $Fe_2O_3$ crystallizes onto the CLI surface. FeCLI appears to be a suitable catalyst for Fenton-like reactions offering advantages, such as activity even at neutral pH, the degradation of different organic pollutants (organic dyes, personal care products, phenol, furfural, etc.) through adsorption and oxidation and recyclability with minimal iron leaching [34–38].

The preparation of MOCLI catalysts very often requires pretreatment of the CLI and its transformation to HCLI. HCLI has a larger specific surface area and possesses acidity. Thus, photocatalitically active $SnO_2$-containing CLI was prepared through the conversion of CLI to HCLI, treatment of HCLI with $SnCl_2$ in an alkaline media and calcination of the product to $SnO_2$-CLI [28].

MOCLI can also be obtained via precipitation, impregnation, hydrothermal crystallization, sol-gel and solid-state dispersion [39–43]. The catalytic performance of MOCLI shows dependance on the modification parameters, such as the metal concentration, reaction temperature, acidity, calcination temperature, etc. As an example, the photocatalytic activity of Cr-doped $TiO_2$CLI strongly depends on the calcination temperature, which decreases with an increase in the calcination temperature [39]. Ullah et al. prepared $TiO_2$CLI photocatalysts with different $TiO_2$ crystal phases and particle sizes. The photocatalysts with anatase show better activity due to the appropriate crystal size, surface area and crystal distribution [42]. Ullah et al. also compared the photocatalytic performance of $TiO_2$CLI prepared using different procedures: sol-gel, hydrothermal and in situ hydrothermal methods [43]. The best dispersion of $TiO_2$ particles was obtained with the in situ hydrothermal method due to the preservation of the CLI crystal structure and the smallest extent of particle agglomeration.

In contrast, a partially distorted CLI and aggregates of $TiO_2$ were obtained using sol–gel and hydrothermal treatments. These are the main reasons for lower catalytic activity.

Since the acidity of a catalyst plays a crucial role in many catalytic reactions, many studies have been focused on HCLI preparation. Two treatments proved to be suitable: (a) conversion of CLI to $NH_4$-CLI, in which an ion-exchange reaction mediated by $NH_4^+$ ions, followed by the calcination of $NH_4$-CLI at about 500 °C, gives HCLI and (b) the direct conversion of CLI to HCLI using an acid treatment. Most commonly, strong mineral acids, such as HCl, $HNO_3$ or $H_2SO_4$ are used. To minimize the loss of crystallinity of the CLI lattice, mild treatment conditions are preferable [44]. In the acid treatment, exchangeable cations from the CLI lattice are replaced by hydronium ions, which is accompanied by partial dealumination of the lattice (Figure 4). Al leaching leads to the formation of lattice vacancies and the appearance of extra-framework aluminum species (octahedrally and five-coordinated Al species) present in the pore system [44,45]. These species create strong acid Lewis sites, in contrast to Brönsted acid sites, which are formed by bridging hydroxyl groups (Si–(OH)–Al).

**Figure 4.** Schematic illustration of the dealumination and desilication of clinoptilolite.

The acid treatment of CLI improves both porosity and catalytic performance. Raw CLI has a relatively low specific surface area (up to 40 $m^2$ $g^{-1}$), as well as micropore volumes. Considerable increases in the specific surface area (even five times), micropore volume and external surface area have been observed during the conversion CLI to HCLI [15,44,46]. This is ascribed to partial dealumination of the aluminosilicate lattice, resulting in opening of the pore system. Also, almost complete replacement of the metal cations by $H^+$ cations occurs and contributes to more free space within the CLI lattice [44,45].

The acidity of CLI can also be also through desilication, which includes the partial removal of $Si^{4+}$ during alkaline treatment (usually by applying NaOH). Si removal can occur at different extents, from about 0.5 wt.% (using 0.05 mol NaOH $dm^{-3}$) to 7.6 wt.% (by 0.8 mol NaOH $dm^{-3}$) without damage to the crystal structure. A higher degree of desilication causes a partial loss of the CLI crystallinity [47–50]. It is worth noting that desilication does not lead to a significant increase in the CLI specific surface area in comparison to

dealumination [47]. Desilication contributes significantly to the acidity increasing both the Brönsted and Lewis acidity [49].

### 2.2. Catalytic Performance of Clinoptilolite

The availability and structural features of CLI result in its widespread application in many areas, including various industrial processes, agriculture, veterinary fields, medicine and environmental protection [51–57].

Moreover, the possibility of the chemical modification of the CLI and its thermal stability make CLI is a good candidate for catalytic applications. CLI shows catalytic activity in the degradation of different environmental pollutants, including organic dyes, pharmaceuticals, herbicides, pesticides, $NO_X$, etc. [28,40,58–60]. Nowadays, CLI has been also applied in biomass conversion to biofuels, biofuel additives and important industrial chemicals [29,61,62].

#### 2.2.1. Degradation of Organic Dyes

The presence of organic dyes in wastewater and their degradation are important issues in environmental protection. They are extensively used in textile, paper, food and cosmetic industries. The complex structures of organic dyes and their chemical stability make degradation very complicated. Additionally, most of these organics are toxic and carcinogenic and cause considerable environmental damage and seriously affect human health [63–65]. The focus of the latest research is finding novel and environmentally friendly materials applicable to photocatalysis, which is one of the advanced oxidation processes usable for the degradation of organic dyes. In these studies, the CLI also finds a significant role.

CLI has shown to be good support for the photocatalytically active particles of metal oxide (MO), such as $TiO_2$, ZnO, $SnO_2$, CuO, NiO, etc. [28,42,66,67]. It has been found that the adsorption affinity of the CLI towards organic dyes brings more organic dye molecules near the catalyst surface where the production of hydroxyl radicals occurs. This leads to the creation of many active sites for the adsorption of intermediates and performance of catalytic reactions. The obtained results revealed an important role of CLI, which enhances the activity of MO particles. The enhanced reactivity of MO-CLI is ascribed to a synergistic effect between the MO particles and the CLI lattice. The CLI lattice prevents the aggregation of MO particles by fixing them onto ion-exchange sites and also enables electron-hole recombination, which both contribute to the photocatalytic reaction [28,66,68,69].

It has been proven that the photocatalytic performance of the MO-CLI depends on the type of dyes and MO and irradiation source, but also the reaction conditions, such as the pH [28,56,68]. The selected literature data are given in Table 1. It can be noticed that CuO-CLI is applicable in the degradation of both methylene blue and bromophenol blue. Better efficiency is observed for methylene blue due to the fact that highly stable benzylic radicals are formed by attacking hydroxyl radicals on the dye molecules [68]. The significant role of CLI was demonstrated for ZnO-CLI showing that an increase in the ZnO concentration onto the CLI surface decreases its photodegradation activity toward bromothymol blue due to the aggregation of ZnO particles, decrease in the effective surface area and collision of ZnO with free dye molecules [69]. A similar effect was also found for $SnO_2$–CLI [28].

The method of preparation of MO–CLI is also important since it affects MO particle sizes and their aggregation. An in situ hydrothermal method leads to the good dispersion of $TiO_2$ particles onto the CLI surface, whereas aggregated $TiO_2$ on a partially distorted CLI structure is reported for sol–gel preparation [43]. Consequently, catalysts synthetized through in situ hydrothermal methods show a high catalytic performance with a removal efficiency of 98% in 180 min in the degradation of crystal violet dye under UV irradiation.

**Table 1.** CLI-based catalysts for the photocatalytic degradation of different organic dyes.

| Catalyst | Dye | Irradiation Source | Experimental Conditions | Degradation Efficiency | Reusability | Ref. |
|---|---|---|---|---|---|---|
| TiO$_2$-CLI | Reactive Black 5 | 8 W UV lamp | 0.4 g dm$^{-3}$, 10 ppm, pH = 6, 500 min | 86% | / | [66] |
| TiO$_2$-CLI | Acid orange 7 | 30 W UV-C Hg lamp | 80 g dm$^{-3}$, 10 ppm, 90 min | 85% | 67% after four cycles | [70] |
| TiO$_2$-CLI | Crystal violet | 20 W UV-C Hg lamp | 0.5 g dm$^{-3}$, 10 ppm, pH = 6, 180 min | 51% | / | [42] |
| Fe-TiO$_2$-CLI | Methylene blue | UV irradiation | 1 g dm$^{-3}$, 1·10$^{-6}$ M, pH = 5.4, 30 min | 100% | | [60] |
| ZnO-CLI | Bromothymol blue | 35 W Hg lamp | 0.1 g dm$^{-3}$, 4 ppm, pH = 7.8, 300 min | 80% | 48% after three cycles | [69] |
| CuO-CLI | Methylene blue | 75 W Hg lamp | 0.2 g dm$^{-3}$, 7 ppm, pH = 5.9, 180 min | 61% | 28% after four cycles | [68] |
| CuO-CLI | Bromophenol blue | | | 32% | 17% after four cycles | |
| SnO$_2$-CLI | Methylene blue | Visible light lamp, 8 mW cm$^{-2}$ | 0.2 g dm$^{-3}$, 10 ppm, pH = 6, 180 min | 45% | 30% after three cycles | [28] |
| CLI | Rhodamine B | 100 W LED cool daylight lamp | 0.75 g dm$^{-3}$, 4.8 ppm, 600 min | 70% | 60% after three cycles | [71] |
| CLI | Methylene blue | Visible light lamp, 8 mW cm$^{-2}$ | | | | [56] |

The complete photodegradation of methylene blue under UV light in 30–45 min was achieved for both Fe– and Cd–TiO$_2$–CLI. The catalysts were prepared by doping Fe(III) and Cd(II) into TiO$_2$-CLI. The doping generates shallow charge traps in the zeolite crystal structure decreasing the recombination rate of the electron hole pairs and excitability by visible light [60].

The reusability of the CLI-based catalysts has also been studied as a very important issue for the catalyst application. Through thermal treatment (usually up to 300–400 °C), most of zeolite-based catalysts can be regenerated and used for new catalytic performance [26,28,56,69,70]. A partial loss of the photocatalytic activity has been noticed and ascribed to the following: (a) surface blockage by degradation products and/or a partial blockage of the pore system and (b) partial leaching of MO species.

Interestingly, pure CLI also exhibits photocatalytic activity in the degradation of organic dyes, such as rhodamine B under visible light [71]. Prior to the photocatalytic performance, both CLI and HCLI were mechanochemically treated in water and in air. The results showed the following: (a) preservation of the crystal structure only under milling in water, (b) a partial dealumination and formation of an extended meso-macroporous structure and c) enrichment of the surface of CLI and HCLI with silica and Fe(III), as well as an increase in the surface content of OH-groups. Photocatalytic activity was ascribed to the presence of Ti and/or Fe species, which are usually found in small amounts as impurities in zeolitic tuffs. In addition, the Fe–O species, as active sites, have also been proposed as photocatalytic active centers [66].

Very recently, the results of the photocatalytic degradation of the cationic dye methylene blue under environmentally friendly conditions (visible light, atmospheric pressure and room temperature) and in the presence CLI tuffs from several deposits (Serbia, Turkey, Iran, Romania and Slovakia) have been reported [56]. The results show the total degradation of methylene blue, which is attributed to a joint effect of the initial adsorption and

subsequent degradation under irradiation with visible light. It has also been shown that the photodegradation efficiency depends on the origin of the CLI tuffs and that the photocatalytic activity increases with an increase in the iron content in the tuff. This suggested that the Fe species can be responsible for the photocatalytic degradation of methylene blue. For all studied zeolitic tuffs, the best activity was observed at pH = 6 reaching total degradation between 70 and 91% ($C_0$ = 10 mg dm$^{-3}$, 0.2 g dm$^{-3}$ of photocatalyst, over 300 min). Moreover, the recyclability tests using thermal treatment demonstrated good stability and efficiency of the tested CLI.

Besides photocatalytic activity in the degradation of organic dye, the applicability of CLI in the photodegradation of other environmental pollutants, such as xanthate (extensively used and the most effective flotation collector in the mineral processing industry) has been observed. Due to high toxicity and stability of xanthate and its derivatives, catalysts based on CLI could have a promising role in environmental protection. TiO$_2$-CLI synthetized using a hydrothermal method was tested in the degradation of sodium isopropyl xanthate under UV light irradiation [41]. Over 90% of the pollutant was degraded within 30 min under neutral conditions. The efficiency of the catalyst was attributed to the synergistic effect between the CLI lattice and TiO$_2$ particles. Additionally, an MoS$_2$/TiO$_2$-CLI catalyst with a hierarchical structure showed enhanced photodegradation activity towards the common Na-based xanthates, such as sodium ethyl xanthate, sodium isopropyl xanthate, sodium butyl xanthate and sodium isoamyl xanthate [72]. This was attributed to the excellent ability of CLI to support the formation of MoS$_2$/TiO$_2$ and heterojunctions within the catalysts. The degradation efficiency gradually increased with the increase in the xanthate molecular weight, reaching over 90% for sodium isopropyl xanthate.

High photocatalytic activity of TiO$_2$-CLI has been reported in the degradation of terephthalic acid, which is a raw compound used for the production of polyester fibers and films, under UVC illumination [73]. TiO$_2$-CLI was found to be the most photoactive catalyst reaching a degradation rate of 94% by using 0.75 g dm$^{-3}$ of catalyst and at an initial concentration of terephthalic acid of 20 mg dm$^{-3}$.

Catalysts based on CLI have also been applied in the photodegradation of agricultural pollutants. TiO$_2$-CLI has been employed in the photocatalytic degradation of two widely used herbicides 2,4-dichlorophenoxyacetic acid (2,4-D) and a 2-methyl-4-chlorophenoxyacetic acid (MCPA) mixture, under ultraviolet and sunlight irradiation [59]. Although the degradation efficiency under sunlight was lower than that obtained using UV irradiation (35% for MCPA and 30% for 2,4-D), such catalytic systems have been reported to be acceptable for use under sunlight. The results showed that the CLI lattice helps the generated electrons to be homogenously distributed throughout the catalyst and prevent or limit the electron hole recombination.

Furthermore, it has been found that CLI is active in the photodegradation of herbicides based on aniline and its derivatives. FeO-CLI showed good photocatalytic activity in the degradation of 2,4–dichloroaniline [74]. Under optimal reaction conditions (13 wt.% of FeO and catalyst dose of 0.5 g dm$^{-3}$, initial herbicide concentration of 15 mg dm$^{-3}$ and pH = 5), a degradation rate of 82% was achieved after 300 min under irradiation using a 30 W W-lamp. This was explained by the good dispersion of FeO on the ion-exchange sites of the CLI lattice, which prevents aggregation, as well as the reduction of the e$^-$/h$^+$ recombination process. Reused FeO-CLI showed good stability through three reuse cycles, although catalysts were regenerated at high temperatures (270 °C). Additionally, the photocatalytic degradation of dichloroaniline was tested using CuO-CLI [75]. A degradation rate of 90% over 300 min was achieved using the catalyst with a 3.9 wt.% of CuO at pH = 3. The photocatalytic activity was attributed to the joint effects of prevention of the aggregation of CuO particles and the absence of the e$^-$/h$^+$ recombination process.

### 2.2.2. Degradation of Pharmaceuticals

Due to their low biodegradability, high persistence and bio-accumulation, pharmaceuticals are recognized as emerging pollutants that pose a severe threat to the environment

and risk to human health. Population growth and the uncontrolled consumption of pharmaceuticals lead to an increase in pharmaceuticals and their related metabolites in the environment. They are mainly released in pharmaceutical industry wastewater and domestic and hospital wastewater. Pharmaceuticals are detected in concentrations ranging from ng dm$^{-3}$ to μg dm$^{-3}$. Due to their high resistance, pharmaceuticals cannot be removed using conventional technologies, and there is a necessity to develop novel materials and technologies for their efficient degradation in water ecosystems. CLI has been identified as a prospective environmental material that can be applicable for the design of different catalytic systems to be used in the degradation of pharmaceuticals.

MO-CLI has been studied in the degradation of different types of pharmaceuticals, including the most frequently used antibiotics, beta-blockers, diuretics, antihistamines, etc. [40,76–78].

NiO-CLI has been tested for the degradation of antibiotics, such as cephalexin and cotrimaxazole, under Hg lamp irradiation [76,77]. The degradation rate of both antibiotics was considerably increased by supporting nano NiO onto the CLI. Under optimal reaction conditions (0.2 g dm$^{-3}$ of the photocatalyst, pH = 4.5 and irradiation time of 300 min), about 76% of cephalexin was degraded. Regeneration of the photocatalyst at 750 °C partially decreased the activity, but it remained at 72% of its initial activity after three reaction cycles. Under similar reaction conditions (0.2 g dm$^{-3}$ of the photocatalyst, pH = 3 and irradiation time of 300 min) about 40% of cotrimaxazole was degraded. It has been suggested that the adsorption properties of the CLI enhanced the chance of hydroxyl radicals attacking the adsorbed antibiotic molecules and led to faster degradation.

FeO-CLI was tested for the photocatalytic degradation of cotrimaxazole, under irradiation with an Hg lamp [79]. FeO particles immobilized onto nano particles of CLI have been found to be responsible for the photocatalytic activity. The results demonstrated the importance of a CLI support for the adsorption of cotrimaxazole and also for the immobilization of FeO particles, which increases their availability for photons. The degradation rate was affected by the reaction conditions, reaching the highest value when using 0.5 g dm$^{-3}$ of the photocatalyst, at pH = 4.3, over 300 min. The photocatalyst retained about 60% of its initial activity after three repeated reaction cycles.

ZnO-CLI has been studied for the degradation of furosemide, one of the most widely used diuretics [40]. The catalyst was synthetized via sonoprecipitation. Discreet, non-agglomerated ZnO nanoparticles were obtained at the CLI surface in different concentrations (5–20 wt.%). The specific surface was increased twice in comparison to ZnO-CLI obtained via conventional precipitation. The degradation of furosemide is significantly affected by pH, showing the highest rate at a pH of about 7. About 86% of furosemide was degraded over 90 min under optimal reaction conditions: catalyst amount of 0.75 g dm$^{-3}$, $C_0$ = 10 mg dm$^{-3}$, pH = 7 under UVA irradiation. The degradation of furosemide follows a Langmuir–Hinshelwood kinetic model, with $k_{app}$ about twice as high as that observed for free ZnO. This clearly supports the important role of CLI in the degradation process. The catalyst reached about 94% of its initial activity after five reaction cycles showing good stability.

The role of CLI has also been proven in the photodegradation of beta-blockers, such as metoprolol [80]. Compared to free TiO$_2$, the superior photocatalytic degradation of metoprolol at all the studied pHs (from 3 to 9) was achieved for TiO$_2$-CLI. Total degradation was achieved at pH = 3, under irradiation with near-UV light over 360 min (1 g dm$^{-3}$ of catalyst and $C_0$ = 20 mg dm$^{-3}$) due to the favorable adsorption of hydroxyl ions on the TiO$_2$-CLI surface and the formation of hydroxyl radicals that are responsible for the degradation process. The role of CLI in this catalytic system was found to not only be in the prevention of the agglomeration of TiO$_2$ but also in the stabilization of radical cations photogenerated on the surface of immobilized TiO$_2$. Also, the CLI lattice is marked as an electron trapper at the CLI-TiO$_2$ interface, which prevents the charge recombination.

The photodegradation performance can be enhanced by coupling two or more semiconductors. Thus, the photocatalytic efficiency of the photocatalysts TiO$_2$/Fe$_2$O$_3$ and

ZnO/Fe$_2$O$_3$–CLI in the degradation of frequently used antihistamines–diphenhydramines has been confirmed [78]. Fe$_2$O$_3$ was selected due to its role in coupling semiconductors, in increasing ZnO activity, as well as in extension of the optical absorption edge to the visible light region for TiO$_2$. The ZnO/Fe$_2$O$_3$-CLI also showed a higher photocatalytic performance compared to TiO$_2$/Fe$_2$O$_3$-CLI under irradiation with UV light. Under the optimal reaction conditions, the degradation of diphenhydramine mediated by ZnO/Fe$_2$O$_3$-CLI was 95% ($C_0$ = 50 mg dm$^{-3}$, hydrogen peroxide 50 mg dm$^{-3}$, 100 min, photocatalyst amount 0.5 g dm$^{-3}$, pH = 10) and for TiO$_2$/Fe$_2$O$_3$-CLI, it was 80% (50 mg dm$^{-3}$, hydrogen peroxide 50 mg dm$^{-3}$, 120 min, photocatalyst amount 0.5 g dm$^{-3}$, pH = 5). The role of CLI in the design of photocatalysts is reflected in the recovery ability and reuse of the catalysts five times.

Nanoparticles of PbS and CdS immobilized onto magnetic Fe$_3$O$_4$-CLI show photocatalytic activity in the photodegradation of the antibiotic cefotaxime under illumination with a medium-pressure Hg lamp [81]. It has been suggested that the Al–O and Si–O bonds in the CLI lattice show semi-semiconductor characteristics due to the formation of e$^-$/h$^+$ pairs under illumination and decrease the photodegradation efficiency of the CLI. Magnetic Fe$_3$O$_4$-CLI exhibits slightly higher efficiency than the parent CLI due to the semiconductor properties of the Fe$_3$O$_4$. Moreover, the photocatalytic activity of the CdS particles immobilized at the CLI surface was higher than that for free particles, which was explained by their homogenous dispersion. The highest photodegradation efficiency was achieved for coupled CdS/PbS immobilized onto CLI, which was attributed to the CLI role in the prevention of particle agglomeration and the existence of permanent internal electric fields in the CLI lattice. This is the driving force in the separation of the photoinduced e$^-$/h$^+$ pairs in CdS. Under optimal reaction conditions ($C_0$ = 5 mg dm$^{-3}$, 250 min, photocatalyst amount 0.7 g dm$^{-3}$, pH = 6), the degradation efficiency was about 84%. Photocatalyst activity was preserved after six reaction cycles.

### 2.2.3. Biomass Conversion

Much effort is being put into using renewable resources to produce industrially valuable chemicals. Great benefits are expected from the conversion of biomass to industrial chemicals, materials and energy products. Biomass is any organic product that originates from plants and animals. It is an ideal, abundantly available, inexpensive and renewable resource. Lignocellulose is a very abundant biomass. It is primarily composed of hemicellulose, cellulose and lignin. The last one is the most abundant.

The conversion of biomass into valuable products is complex and requires processes that are environmentally friendly. Different kinds of natural materials, such as mesoporous solids, clays, metal oxides, organic resins and zeolites have been studied as catalysts in the conversion of biomass [29,61,62,82,83].

The thermal treatment of lignocellulose in the absence of oxygen (pyrolysis) is a very convenient process for bio-oil production. Due to its corrosive nature and low heating value, the high water and oxygen content of the obtained bio-oils requires the upgrading of bio-oils to make them compatible with conventional fuels. CLI has been employed in the catalytic pyrolysis of rice straw in a fixed-bed reactor at 550 °C [84]. The physical characteristics of the bio-oil obtained, such as the density, flash points and pour points were improved for previously acid-treated rice straw. The use of the CLI improves the selectivity of the most valuable aromatic hydrocarbons (benzene, toluene and xylene) from 7.7%, 13% and 8.7% for raw to 10.5%, 15.8% and 21.05% for the acid-washed biomass, respectively. The high heating value was increased from 19.5 to 26.3 MJ kg$^{-1}$ as a result of an increase in hydrocarbons and the reduction of oxygenates in the bio-oil through catalytic reforming. Although raw CLI showed promising results in the catalytic upgradation, dealumination and MO immobilization make CLI more active. Synergism between MO particles and the CLI framework enhances the catalytic activity. Fe$_2$O$_3$, ZnO and CuO–CLI showed good catalytic performance in the hydrodeoxygenation of bio-oil refining [85]. All refined bio-oils indicated upgraded physicochemical properties with high heating values that ranged from

17 to 21 MJ kg$^{-1}$. The ZnO-CLI (5 wt.%) catalyst showed the best catalytic performance at 275 °C with a 50% conversion rate. Differences in the catalytic activity of MO-CLI were related to the textural properties of the catalysts, which influence reaction pathways.

Esterification and transesterification between free fatty acids (or triglycerides) and alcohols are used for biodiesel production. Both edible and non-edible oils, algae and bacterial biomass are promising renewable resources for biodiesel production. Methanol and ethanol are the mostly used alcohols due to their polarity, availability, low cost and superior reactivity. The use of solid catalysts offers a higher tolerance of free fatty acids and water content in the feedstock, mitigates the soap formation and provides catalyst reusability. Using CLI-based catalysts and calcined industrial phosphoric waste, biodiesel was produced via the transesterification of waste cooking oil [62]. The highest purity of biodiesel (around 85%) was obtained by using an oil-to-methanol volume ratio 1.47, a catalyst dosage of 8.08 wt.%, at 54.72 °C, and a duration of 119 min. During catalytic cycles, partial deactivation occurs (up to 20%).

$SnO_2^-$ and sulfated $SnO_2$-CLI showed high catalytic activities in the esterification of levulinic acid, one of the platform chemicals obtained from biomass, to octyl and ethyl levulinate [29]. These esters are of great interest due to their application as fuel additives, solvents and plasticizers. Octyl levulinate is a suitable replacement for synthetic and mineral-oil based lubricants. Total conversion was achieved for sulfated $SnO_2$-CLI, which was attributed to the presence of a high number of Brönsted and Lewis acid sites. Most importantly, the catalytic activity remains stable, which was confirmed in five repeated reaction cycles. This showed that the leaching of sulfate groups (which is the main disadvantage of a $SO_4$-$SnO_2$ super acid) was preserved, which can be attributed to the structural features of the CLI lattice. The catalysts showed higher catalytic activity in the esterification of levulinic acid with octanol than with ethanol due to a higher coke formation in the latter process [29].

The catalyst prepared through the immobilization of phosphomolybdic acid (40%) on the $Fe_3O_4$-CLI was used for biodiesel production from *salvia mirzayanii* as a new non-edible source via electrolysis [61]. The produced biodiesel shows similar physicochemical properties to the diesel. The highest yield (80%) was achieved at operating conditions of 0.5 wt.% catalyst and a methanol:oil ratio of 12:1, at 75 °C, over 8 h. The magnetic CLI offers a high specific area and oxygenated functional groups that contribute to a uniform distribution and immobilized phosphomolybdic acid molecules as active sites. No significant decrease in biodiesel production was observed during the four repeated cycles.

It is worth mentioning that one of the most produced and potential byproducts of transesterification is glycerol. Alkaline-treated CLI showed activity in the catalytic acetalization of glycerol to fuel additives, such as Solketal [86]. Catalytic tests performed in a glass tube reactor at 50 °C for 90 min indicated a significant increase in glycerol conversion (90.4%) compared to parent CLI (3.5%). This was ascribed to a change in the textural properties of the CLI and to improvements in the diffusional limitation of glycerol molecules.

In addition, the catalytic performance of acidic CLI was shown in the esterification of glycerol with different $C_1$–$C_4$ alcohols to corresponding ethers, which are useful components of new sustainable synthetic fuels [87]. The tested catalysts are highly active, especially with alcohol containing the branched, tertiary alkyl groups due to the enhanced stability of the intermediate formed carbocations. A total conversion of 78% to mono and di ether were obtained with tert-buthanol at 140 °C after 4 h giving selectivity of 75% and 25%, respectively. Notably, the catalyst exhibited good stability with only a moderate activity loss (glycerol conversion decreased from 74% to 57% after the fourth run).

Isomerization reactions alter the molecular geometry, such as the typical transformation of hydrocarbons with a straight chain into branched-chain hydrocarbons. The reaction is involved in the biomass-conversion processes, and it is catalyzed by different types of acid catalysts.

CLI shows activity in the isomerization of $\alpha$-pinene (a terpene extracted from biomass) into industrially important compounds, such as camphene and limonene, which are especially important due to their versatile applications, including pharmaceutical, food and cosmetic industries [88]. The most active CLI was obtained by treatment with 0.1 M $H_2SO_4$, showing activity even at a low temperature (30 °C). A 100% conversion rate was achieved at 70 °C after only 4 min. Camphene and limonene were obtained with the highest selectivity (50% and 30%, respectively). It has been shown that the catalytic activity of CLI is influenced by the treatment due to changes in the textural properties and the acid site concentration and distribution. A similar conclusion was derived for the activity of dealuminated CLI obtained through treatment with HCl with different concentrations (0.05–11.5 M), as well as with calcined $NH_4$-CLI [15]. The most active catalyst was obtained using CLI treated with 0.05 M HCl: conversion rate of 41% at 90 °C after 180 min, and selectivity to camphene and limonene was 57% and 32%, respectively.

The CLI calcined at different temperatures was also tested for the isomerization of $\alpha$-pinene [89]. A decrease in the catalytic activity with an increasing calcination temperature was observed due to a partial loss of acid sites via calcination. Total conversion was achieved at 155 °C over 180 min, with selectivity of 39% and 25% towards camphene and limonene, respectively.

The catalytic performance of the $Cr^{3+}$- and $Fe^{3+}$-loaded CLI was studied in the liquid-phase isomerization of $\alpha$-pinene [90]. The major reaction products were camphene and limonene. The conversion rate was strongly dependent on the type and amount of metal cations. The highest conversion rate (about 94%) was obtained for Fe(III)-CLI (with approx. 0.7% of Fe) at 155 °C over 480 min, with selectivity of 65.6% and 4.4% for camphene and limonene, respectively. The activity was ascribed to an increase in Brönsted acid sites. In addition, structural stability of the catalyst was observed during reuse tests: conversion rate of $\alpha$-pinene was in the range 85%–95%, and a nearly constant yield of camphene (40%–46%) was achieved.

The isomerization of limonene over CLI has been reported as a cost-effective and environmentally friendly process due to the fact that orange peel waste was used to produce terpinolene and *p*-cymene, products with versatile applications in food, cosmetic and pharmaceutical industries [91]. The main reaction parameters affecting limonene isomerization were the temperature, reaction time and catalyst content, reaching the greatest yield of terpinolene (about 39 mol.%) after a reaction for 60 min at 175°C, using 10 wt.% of catalyst. However, the production of *p*-cymene using similar reaction conditions as for terpinolene was significantly slower (1440 min).

CLI has been tested for the conversion of dihydroxyacetone (DHA) to carboxylic acids, such as lactic, formic or pyruvic acid [92,93]. These acids are important and widely used in many industrial areas. To achieve catalytic activity, the CLI was subjected to the following: (a) an ammonium exchange reaction, (b) hierarchization in an acidic environment and (c) ion exchange with iron, zinc or cobalt ions in hydrothermal conditions with the use of ultrasound. An almost five-fold increase in the specific surface area was observed after acid treatment, whereas the use of a sonication technique requires less time and eliminates the use of solvents. Catalytic reactions were performed using a one-pot method in an aqueous medium to provide a more economical and environmental system. Depending on the metal type, the conversion of DHA resulted in the production of carboxylic acids (lactic acid, formic acid, pyruvic acid, acetic acid and levulinic acid) with different selectivities but with 100% transformation of the raw material. The highest lactic acid yield (66.2%) was achieved with Co-CLI, formic acid (93.6%) with Cu-CLI and 87.4% acetic acid with Fe-CLI.

### 2.2.4. Conversion of NOx

Global energy consumption has increased the emission of nitrogen oxides (NOx; mainly NO and $NO_2$), leading to serious environmental and health problems (acid rains photochemical smog, ozone depletion). The selective catalytic reduction of NOx (SCR) is one of the most widely applied technologies for NOx reduction.

Catalysts for SCR must be highly active at low temperatures and have good resistance to $H_2O$ and $SO_2$. Various zeolites, such as BEA, FAU or ZSM, have been tested [94–97]. CLI has also been recognized as a suitable candidate for SCR using ammonia or hydrocarbons as reducing agents [98–108]. Even non-modified CLI possesses catalytical activity, which was ascribed to the presence of iron species, which are the main impurity in the zeolitic tuffs [98–100].

The catalytic performance can be enhanced through the immobilization of active metal oxide particles at the CLI surface. Iron oxides were mostly studied because of their outstanding thermal stability, environmental compatibility, weak oxidizing properties and high tolerance to the presence of SOx in flue gas [98,99,106,107]. Saramok et al. studied $Fe_xO_y$-CLI in $NH_3$-SCR in a nitric acid plant [98,99]. The used catalyst was prepared by converting CLI to HCLI and then through the precipitation of Fe(III) onto the CLI surface. The catalytic tests were carried out using exhaust gases from the nitric acid pilot plant at a constant hourly gas space velocity of $26,500^{-1}$ in the temperature range of 150–450 °C. The catalytic results showed high activity of the CLI catalyst (conversion rate of 58% was achieved at 450 °C).

In addition, it was shown that the activity depends on the form in which the catalyst is used. The pelletized catalyst gives significantly better results than powdery ones. The catalytic efficiency was quite satisfactory under industrial conditions at 450 °C and maintained a very favorable selectivity towards $N_2$. Iron content also affects the catalytic performance of the pelletized catalyst. Saramok et al. [99] reported a conversion rate of 90% at 350 °C for the catalyst with a 8.5–12 wt.% of Fe. The high activity was attributed to a variety of iron species, including isolated Fe(III) ions and FexOy oligomers. FeCLI showed superior catalytic performance compared to the natural layered clays bentonite and vermiculite [107]. The lowest formation of reaction by-products was attributed to the presence of well dispersed isolated Fe(III) cations inside the CLI lattice and the well-developed pore system of CLI, which both facilitate the access of gas molecules to the active species.

Cu(II)-exchanged CLI was also an efficient catalyst for the SCR with ammonia in an oxygen-rich atmosphere. Conversion of 95% was reported at a relatively low temperature (180–380 °C) with a high selectivity to $N_2$ and satisfactory water tolerance [104].

Zn-exchanged CLI was also active with ammonia as a reducing agent in the presence of excess oxygen at 400–500 °C [105]. Moreover, CLI-based catalysts showed a high catalytic performance in the SCR using hydrocarbons, such as propane, as a reducing agent. HCLI with a mesh size ranging from 20 to 70 showed a high conversion rate (72%) at 450 °C. This was attributed to the absence of diffusion resistance in the pore system and the dominance of the intrinsic reaction rate. The suggested mechanism includes the adsorption of NO and $NO_2$ molecules on HCLI and their reaction with Brönsted acid sites to nitrosonium ions. These ions interact with lattice oxygen atoms, as well as with oxygen originating from the gas phase, producing nitrate species, which are then reduced to $N_2$ by propane.

Recently, catalysts prepared through the coprecipitation of hydrotalcite (HT) on CLI were tested in the SCR reaction with ammonia [108]. The deposition of HT on the CLI resulted in a high conversion rate at 250–450 °C (95%). This was attributed to a synergetic effect of Brönsted acid sites in CLI (which allowed for efficient $NH_3$ adsorption) and to the presence of Lewis acid sites on to the HT.

$MnO_2$-CLI is also active for SCR. The activity was affected by the Mn content, showing an increase in propane-NOx conversion by increasing the $MnO_2$ content from 0.05 to 0.2 wt.% [101]. A further increase in the $MnO_2$ content attenuates the conversion due to a reduction of the specific surface area. The maximum NOx conversion (71%) was achieved at 150 °C with a catalyst loaded with 0.2 wt.% $MnO_2$.

## 3. Conclusions

The production of novel chemicals and materials, industrially valuable chemicals and energy products in accordance with environmental requirements are given high priority today. Natural clinoptilolite is acknowledged as a potential candidate to be used in various

catalyzed processes in accordance with the data presented. Its prevalence and high thermal and chemical stability can be attributed to this.

The potential for incorporating various metal species into the clinoptilolite lattice and/or partial dealumination of the clinoptilolite surface both enhance the material catalytic performance. These allow for the design and fine-tuning of clinoptilolite catalytic activity for various applications in environmental protection. Catalytically active species can be dispersed finely at the clinoptilolite surface and be prevented from agglomerating due to the lattice microporosity. The result is that some nano oxide particles are more active when immobilized at the clinoptilolite surface as opposed to when they are free. Furthermore, because they are immobilized, clinoptilolite-based catalysts exhibit excellent reusability and are not subject to leaching or deactivation.

The presented research data clearly show the usefulness of clinoptilolite-based catalysts in wastewater treatment, including the photodegradation of various organic pollutants, such as organic dyes, pharmaceuticals and herbicides, under mild lighting conditions. The conversion of biomass into biofuels, biofuel additives and industrially useful chemicals is another area in which they are active. Finally, clinoptilolite-based catalysts can be effectively used to prevent air pollution. MO-CLI and HCLI catalysts are the most promising candidates. They show high catalytic activity in the conversion of NOx in selective catalytic reduction at a moderate temperature and with a very low amount of byproducts.

According to the presented data, natural clinoptilolite has great potential for the reactions of innovative environmentally friendly catalysts.

**Author Contributions:** Conceptualization and methodology, J.P. and N.R.; investigation, J.P. and N.R.; writing—original draft preparation, J.P.; writing—review and editing, N.R.; visualization, J.P.; supervision, N.R.; project administration, N.R.; funding acquisition, N.R. All authors have read and agreed to the published version of the manuscript.

**Funding:** This research was funded by the Ministry of Education, Science and Technological Development of the Republic of Serbia (Contract No. 451-03-47/2023-01/200287 and 451-03-47/2023-01/200135).

**Data Availability Statement:** The data presented in this study are available on request from the corresponding author.

**Conflicts of Interest:** The authors declare no conflict of interest.

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
