# Peer review of "Clinoptilolite—An Efficient Carrier for Catalytically Active Nano Oxide Particles"

_minerals, doi:10.3390/min13070877_

Round 1

Author Response

  1. The focus of this review is the catalytic activity of the clinoptilolite zeolite, however, it must be focused more on a special topic. For example, the photocatalysis that this zeolite as a support can effect. Change the title of this review to the paper as a support for certain active metals in photocatalysis.

Response: We have changed the title to better refer to the text (Clinoptilolite – an efficient carrier for catalytically active nano oxide particles).

  1. The paragraphs are very long, they should be shortened.

Response: Long paragraphs are shortened.

  1.  Scientific citations must be placed properly.

Response: Improper citations are now corrected.

  1. Of the objectives of the work, consider the possible change in the name of this review.

Response: The title is changed (Clinoptilolite – an efficient carrier for catalytically active nano oxide particles).

5.- In conclusions, consider the role that this zeolite can play as a support for certain inorganic oxides.

Response: We made changes in the Conclusion and emphasized the role of zeolite in catalytic activity.

Reviewer 2 Report

The review summarized the recent advances in the catalytic use of zeolite, the authors focused on the typical kind of clinoptilolite zeolite, and introduced its application in the degradation of organic dyes, pharmaceuticals and biomass conversion. Generally, this manuscript can be accepted for publication despite the relatively simple orgainization of the content. It is suggested that the authors can list the application of clinoptilolite in NO conversion in the field of air pollution treatment, which is also an important catalytic application.

There are some language errors, the authors should check through the text. 

Author Response

The review summarized the recent advances in the catalytic use of zeolite, the authors focused on the typical kind of clinoptilolite zeolite, and introduced its application in the degradation of organic dyes, pharmaceuticals and biomass conversion. Generally, this manuscript can be accepted for publication despite the relatively simple orgainization of the content. It is suggested that the authors can list the application of clinoptilolite in NO conversion in the field of air pollution treatment, which is also an important catalytic application.

Response: We thank the referee for this suggestion. We focused on the Clinoptilolite applications in the areas we researched. Given the importance of the catalytic conversion of NO, we have expanded the text accordingly.

Reviewer 3 Report

This manuscript focused on the catalytic application of clinoptilolite (CLI) and CLI-based catalysts in degrading dyes and pharmaceuticals as well as biomass conversion. The synthesis of CLI-based catalysts and conventional applications of CLI and CLI-based catalysts were not comprehensively described. Thus, the manuscript should be further enriched before acceptance.

1.     The advantages of CLI (structure, specific surface are, pore size, pore volume, etc.) in comparison with other zeolites should be emphasized.

2.     The methods for modification of CLI and synthesis of various CLI-based catalysts should be described in detail, such as 2.1 Modification of CLI, 2.2 Synthesis of CLI-based photocatalysts, 2.3 Synthesis of CLI-based Fenton-like catalysts, etc.

3.     Except for degrading dyes, pharmaceuticals and biomass conversion, the catalytic application of CLI should be further extended.

4.     The catalytic efficiencies between CLI-based catalysts and other zeolites-based catalysts should be compared.

The authors should further polish the manuscript.

Author Response

This manuscript focused on the catalytic application of clinoptilolite (CLI) and CLI-based catalysts in degrading dyes and pharmaceuticals as well as biomass conversion. The synthesis of CLI-based catalysts and conventional applications of CLI and CLI-based catalysts were not comprehensively described. Thus, the manuscript should be further enriched before acceptance.

  1. The advantages of CLI (structure, specific surface are, pore size, pore volume, etc.) in comparison with other zeolites should be emphasized.

Response: The main advantages of CLI are its availability and low price which are emphasized at p. 2. In comparison to other zeolites, it has a smaller specific surface area but it and the pore size can be enlarged by modification. It was written at p. 5.

  1. The methods for modification of CLI and synthesis of various CLI-based catalysts should be described in detail, such as 2.1 Modification of CLI, 2.2 Synthesis of CLI-based photocatalysts, 2.3 Synthesis of CLI-based Fenton-like catalysts, etc.

Response: The methods of modification are now described in detail.

  1. Except for degrading dyes, pharmaceuticals and biomass conversion, the catalytic application of CLI should be further extended.

Response: We now extended the application of CLI in NO conversion which is given at p. 13.

  1. The catalytic efficiencies between CLI-based catalysts and other zeolites-based catalysts should be compared.

Response: It is difficult to compare the efficiency of different types of zeolites in catalyzed reactions, given that the reaction conditions are usually different (temperature, initial concentrations, pH, amount of catalysts etc.). That is why we decided to compare photocatalytic activity of CLI-based catalysts in the degradation of different organic dyes (Table 1).
